# Efficacy of Two Commercially Available Adsorbents to Reduce the Combined Toxic Effects of Dietary Aflatoxins, Fumonisins, and Zearalenone and Their Residues in the Tissues of Weaned Pigs

**DOI:** 10.3390/toxins15110629

**Published:** 2023-10-27

**Authors:** Micheli Midori de Cerqueira Costa Aoyanagi, Fábio Enrique Lemos Budiño, Jog Raj, Marko Vasiljević, Sher Ali, Leandra Naira Zambelli Ramalho, Fernando Silva Ramalho, Carlos Humberto Corassin, Giovana Fumes Ghantous, Carlos Augusto Fernandes de Oliveira

**Affiliations:** 1Department of Food Engineering, Faculty of Animal Science and Food Engineering (FZEA), University of São Paulo (USP), Av. Duque de Caxias Norte, 225, Pirassununga 13635-900, SP, Brazil; micheli.aoyanagi@usp.br (M.M.d.C.C.A.); alisher@usp.br (S.A.); carloscorassin@usp.br (C.H.C.); 2Department of Agriculture and Food Supply of the São Paulo State, Institute of Animal Science and Pastures, Nova Odessa 13460-000, SP, Brazil; fbudino@iz.sp.gov.br; 3Patent Co., DOO., Vlade Ćetkovića 1A, 24211 Mišićevo, Serbia; jog.raj@patent-co.com (J.R.); marko.vasiljevic@patent-co.com (M.V.); 4Department of Pathology and Legal Medicine, School of Medicine at Ribeirão Preto, University of São Paulo (USP), Ribeirão Preto 14040-900, SP, Brazil; lramalho@fmrp.usp.br (L.N.Z.R.); framalho@fmrp.usp.br (F.S.R.); 5Department of Basic Sciences, Faculty of Animal Science and Food Engineering (FZEA), University of São Paulo (USP), Pirassununga 13635-900, SP, Brazil; giovana.fumes@usp.br

**Keywords:** AFB_1_, FB_1_, ZEN, residues, adsorbent, piglets

## Abstract

Mycotoxins present a significant health concern within the animal-feed industry, with profound implications for the pig-farming sector. The objective of this study was to evaluate the efficacy of two commercial adsorbents, an organically modified clinoptilolite (OMC) and a multicomponent mycotoxin detoxifying agent (MMDA), to ameliorate the combined adverse effects of dietary aflatoxins (AFs: sum of AFB_1_, AFB_2_, AFG_1_, and AFG_2_), fumonisins (FBs), and zearalenone (ZEN) at levels of nearly 0.5, 1.0, and 1.0 mg/kg, on a cohort of cross-bred female pigs (*N* = 24). Pigs were randomly allocated into six experimental groups (control, mycotoxins (MTX) alone, MTX + OMC 1.5 kg/ton, MTX + OMC 3.0 kg/ton, MTX + MMDA 1.5 kg/ton, and MTX + MMDA 3.0 kg/ton), each consisting of four individuals, and subjected to a dietary regimen spanning 42 days. The administration of combined AFs, FBs, and ZEN reduced the body-weight gain and increased the relative weight of the liver, while there was no negative influence observed on the serum biochemistry of animals. The supplementation of OMC and MMDA ameliorated the toxic effects, as observed in organ histology, and provided a notable reduction in residual AFs, FBs, and ZEN levels in the liver and kidneys. Moreover, the OMC supplementation was able to reduce the initiation of liver carcinogenesis without any hepatotoxic side effects. These findings demonstrate that the use of OMC and MMDA effectively mitigated the adverse effects of dietary AFs, FBs, and ZEN in piglets. Further studies should explore the long-term protective effects of the studied adsorbent supplementation to optimize mycotoxin management strategies in pig-farming operations.

## 1. Introduction

Mycotoxins are toxic compounds produced by certain fungi species that can grow on crops, food products, and in various environments [1,2,3]. The most common mycotoxins are aflatoxins (AFs), ochratoxin A (OTA), deoxynivalenol (DON), fumonisins (FBs), and zearalenone (ZEN). When consumed or inhaled, these toxic compounds exert harmful effects on humans and animals. The most pronounced effects involve carcinogenicity, mutagenicity, and immunosuppression, depending on the type, dose, and exposure duration to these toxins [4,5,6]. Carcinogenic effects are developed mostly by AFs and FBs [4,5], while immunosuppression is caused mainly by AFs, DON, and OTA, and infertility and endocrine disruption are attributed to ZEN [7,8,9]. The severity of these toxic effects in production animals may vary according to several factors, including the individual’s age, health status, and susceptibility [5,9]. Thus, control strategies for mycotoxins in food and feed are essential for health protection. This can be done by implementing proper food/feed storage and processing methods, as well as regular monitoring and testing of these toxicants in agricultural commodities and animal feed [10].

Pigs are among the most sensitive animal species to mycotoxins, especially to AFs, FBs, and ZEN [11,12]. The high sensitivity in pigs can be attributed to their single-chambered stomach, which facilitates toxin absorption in the gastrointestinal tract, coupled with their high feed consumption relative to body weight [13]. Mycotoxin exposure can lead to a reduction in the growth rate, feed efficiency, and overall performance in pigs [11]. In addition, AFs and DON also lead to immunosuppression disorder, while ZEN induces reproductive problems like infertility and abnormal estrous cycles [7,8,9]. Due to their sensitivity to toxins and the potential economic impact on the swine industry, it is a common practice to closely monitor and manage mycotoxin contamination in their diet. This includes the use of mycotoxin binders or adsorbents in feed, applying quality control measures in feed production, and regular testing of feed ingredients for mycotoxin to reduce their related health risks and other issues in pigs [7,8,9,11].

Adsorbents for mycotoxins are substances or materials that can bind to mycotoxins and reduce their bioavailability, thus preventing their absorption in the gastrointestinal tract and reducing their potential harm when consumed [14,15]. The common types of adsorbents for mycotoxins include bentonite clay, silicates, and zeolites [15]. These adsorbents are commonly used in animal feed to mitigate the risk of mycotoxin contamination. However, the choice of adsorbent depends on the specific mycotoxin contamination issue and the intended application, such as in agriculture or animal husbandry. In this context, in vivo studies are needed to confirm the efficacy of adsorbents for mycotoxins. Minazel Plus^®^ is a product based on organically modified clinoptilolite (OMC) [16], whereas MycoRaid^®^ is a multicomponent mycotoxin detoxifying agent (MMDA) based on specially selected minerals, *Bacillus* sp., yeast cell wall, and herbal extract to remediate the effect of mycotoxins in animals [12]. Therefore, the objectives of this study were: (1) to determine the efficacy of two commercial adsorbents, OMC and MMDA, to ameliorate the toxic effects of dietary AFs, FBs, and ZEN on piglets’ performance and serum chemistries and (2) to determine the efficacy of the adsorbents to reduce residual concentrations of mycotoxins metabolites in the liver and kidneys of piglets fed the combined mycotoxins.

## 2. Results

### 2.1. Growth Performance

The effects of dietary treatments on the growth performance of piglets fed mycotoxin-contaminated rations with or without commercial adsorbents for 42 days are shown in Table 1. Compared with the controls, the body-weight gain of animals receiving a mixture of AFs (sum of AFB_1_, AFB_2_, AFG_1,_ and AFG_2_), FBs, and ZEN were lower (*p* < 0.05), while pigs receiving these mycotoxins in combination with OMC or MMDA at 1.5 or 3.0 kg/ton had values similar (*p* > 0.05) to the controls. However, feed consumption and feed gain did not differ (*p* > 0.05) among treatments.

### 2.2. Serum Biochemistry

The results of serum biochemistry are summarized in Table 2. No significant differences (*p* > 0.05) were observed between the six treatments for total protein (TP), albumin (ALB), serum aspartate aminotransferase (AST), alanine aminotransferase (ALT), or alkaline phosphatase (ALP).

### 2.3. Relative Organ Weights

The individual body weights of pigs at the end of the trial and their respective organ weights were used to calculate the relative weights of the liver, kidneys, uterus, ovarium, and lungs. The results were expressed as g/kg body weight, as presented in Table 3. Livers from pigs fed with the mycotoxin mixture alone or with 1.5 kg/ton MP had higher (*p* < 0.05) relative weights than controls, while the addition of 3.0 kg/ton OMC or MMDA (1.5 or 3.0 kg/ton) reduced the relative weight of the liver after 42 days of dietary exposure to AFs, FBs, and ZEN. The relative weight of the uterus from piglets in treatment B (BD + AFs + FBs + ZEN) was higher (*p* < 0.05) than controls or animals from treatments D (BD + AFs + FBs + ZEN + 3.0 kg/ton OMC) and F (BD + AFs + FBs + ZEN + 3.0 kg/ton MMDA). No significant differences were observed (*p* > 0.05) among the relative weights of kidneys, ovarium, or lungs.

Size measurements were also performed for the vulvas from piglets after 42 days of intoxication, as given in Table 4. Compared with the controls, the width, height, and length of vulvas were higher (*p* < 0.05) in animals from treatment B (mixed mycotoxins only), while the administration of OMC or MMDA reduced the size values, having a maximum effect with 1.5 kg/ton MMDA.

### 2.4. Histopathology

No histopathological changes were observed in the evaluated uterus from any treatment. The histopathological findings in the liver, kidneys, lungs, and ovaries are presented in Figure 1, Figure 2, Figure 3 and Figure 4, respectively. Control animals fed only with basal diet (BD) did not show any histopathological changes in the liver (Figure 1A), kidneys (Figure 2A), lungs (Figure 3A), and ovaries (Figure 4A). However, animals exposed to the mycotoxin mixture developed moderate liver dysplasia (Figure 1B) in 75% of cases, while those receiving the mycotoxin mixture and 1.5 kg/ton OMC showed mild hepatic dysplasia (25% of cases) (Figure 1C); those fed the mycotoxin mixture with 3.0 kg/ton OMC had no liver changes (Figure 1E). Animals that received 1.5 kg/ton MMDA in addition to the mycotoxin mixture showed mild hepatitis (50% of cases) without dysplasia (Figure 1D), while those exposed to the mycotoxin mixture and treated with 3.0 kg/ton MMDA showed moderate hepatitis (50% of cases) without dysplasia (Figure 1F).

Regarding the kidneys, animals exposed to the mycotoxin mixture developed renal glomerular atrophy (25% of cases) (Figure 2B). However, pigs exposed to the mycotoxin mixture and receiving 1.5 kg/ton OMC exhibited no kidney changes (Figure 2C), while animals receiving the mycotoxin mixture and 3.0 kg/ton OMC exhibited only mild renal interstitial inflammation in 25% of cases (Figure 2E). Animals that received 1.5 kg/ton MMDA in addition to the mycotoxin mixture showed renal glomerular atrophy (25% of cases) (Figure 2D), while those exposed to the mycotoxin mixture and treated with 3.0 kg/ton MMDA showed mild renal interstitial inflammation (75% of cases) (Figure 2F).

Lungs from animals exposed to the mycotoxin mixture developed moderate interstitial pneumonitis and mild pulmonary edema in 100 and 25% of cases, respectively (Figure 3B). Pigs fed the mycotoxin mixture plus 1.5 kg/ton OMC showed mild pulmonary edema (50% of cases) without interstitial inflammation (Figure 3C), while those receiving the mycotoxin mixture and 3.0 kg/ton OMC exhibited no lung changes (Figure 3E). Animals that received 1.5 kg/ton MMDA in addition to the mycotoxin mixture showed mild interstitial pneumonitis and pulmonary edema (75 and 25% of cases, respectively) (Figure 3D), while those exposed to the mycotoxin mixture and treated with 3.0 kg/ton MMDA showed pulmonary edema (25% of cases) but without inflammation (Figure 3F). As for the ovaries, animals receiving the mycotoxin mixture showed a reduced oocyte number in 25% of cases (Figure 4B). Pigs exposed to the mycotoxin mixture and receiving 1.5 kg/ton OMC also presented a reduced oocyte number (25% of cases) (Figure 4C), while those fed the mycotoxin mixture and 3.0 kg/ton OMC exhibited no ovarian changes (Figure 4E). Animals that received 1.5 kg/ton MMDA in addition to the mycotoxin mixture showed an increased oocyte number in 25% of cases (Figure 4D), and those receiving the mycotoxin mixture and treated with 3.0 kg/ton MMDA had no changes in the ovarium (Figure 4F).

### 2.5. Mycotoxin Residues in Liver and Kidneys

The residual levels of AFs, FBs, ZEN, and their metabolites (α-ZEL and β-ZEL) in the liver and kidneys are presented in Table 5 and Table 6, respectively. In Table 5, the addition of AFs, FBs, and ZEN to the BD (treatment B) leads to mean levels in the liver for AFM_1_, AFs, FBs, and ZEN of 0.92 ± 0.07, 3.99 ± 0.22, 2.59 ± 0.97, and 23.3 ± 3.7 µg/kg, respectively. ZEN metabolites (α-zeralenol, α-ZEL, and β-zeralenol, β-ZEL) were not detected in any liver sample. Lower levels (*p* < 0.05) of residual FBs and ZEN were observed in the liver from treatments receiving the evaluated adsorbents. Regarding the residual AFs, no quantifiable levels were observed in the livers of pigs in treatments C-F. It suggests that the mycotoxin contamination was effectively reduced or bound in the presence of OMC or MMDA, as indicated by the variable levels of these toxins across treatments.

The results in Table 6 reveal that kidneys from pigs in treatment B, which involved combined AFs, FBs, and ZEN, exhibited mean concentrations of 2.63 ± 0.31, 9.13 ± 0.25, 4.32 ± 0.65, and 41.6 ± 5.7 µg/kg for AFM_1_, AFs, FBs, and ZEN, respectively. However, in treatment C (BD + AFs + FBs + ZEN + 1.5 kg/ton OMC), the levels were below the LOQ for AFM_1_, AFs, α-ZEL, and β-ZEL, while FBs and ZEN were observed at mean levels of 2.98 ± 1.11 and 34.3 ± 3.3 µg/kg, respectively. These results were similar in treatment D (BD + AFs + FBs + ZEN + 3.0 kg/ton OMC), with quantifiable levels of FBs (3.25 ± 1.56) and ZEN (30.5 ± 4.2). In the same order, treatment E (BD + AFs + FBs + ZEN + 1.5 kg/ton MMDA) provided mean levels for AFM_1_ at 1.06 ± 0.20, AFs at 4.25 ± 0.53, FBs at 2.77 ± 1.02, and ZEN at 23.1 ± 2.9 µg/kg. In treatment F (BD + AFs + FBs + ZEN + 3.0 kg/ton MMDA), the mean values were lower for AFM_1_ (0.80 ± 0.11), AFs (1.92 ± 0.24), FBs (2.63 ± 1.32), and ZEN (17.4 ± 2.1 d), if compared with the data for treatment B. The variations in mycotoxin levels in kidneys indicate that OMC or MMDA supplementation has the potential to reduce the residual levels of mycotoxins and associated health risks in piglets.

## 3. Discussion

In this study, the absence of clinical signs in experimental piglets indicates low-to-moderate exposure of piglets to the evaluated mycotoxins and lack of stress due to the high comfort of animals. Moreover, these results indicate that the adsorbents evaluated at 1.5 or 3.0% of the diet did not negatively affect the health status of experimental animals. However, the combined mycotoxins tested negatively affected the body weights of female piglets after 42 days of intoxication (Table 1), although no effects were observed on feed consumption or feed gain. The inclusion of adsorbents OMC or MMDA at 1.5 or 3.0% decreased the negative effects of combined mycotoxin, thus increasing the body weights of piglets during 42 days of intoxication.

One of the most significant economic effects of mycotoxicosis in pig production is the growth-rate reduction [17]. In this study, a mixture of AFs, FBs, and ZEN was used, leading to combined toxic effects on the exposed animals. The effects of simultaneous exposure to multimycotoxins are complex and may be classified into synergic, additive, and antagonist categories [18]. Previous studies have identified some possible synergistic and additive interactions of co-occurring mycotoxins, such as AFs and FBs associated with reduced body-weight gain [19,20], which agrees with the data reported here. Changes in protein synthesis, gene expression, and enzyme kinetics are considered the main mechanisms by which mycotoxins impair piglets’ performance [21].

As for specific toxic effects, AFB_1_ is well-known for both its carcinogenic and teratogenic properties [22,23]. Hepatic microsomal cytochrome P450 (CYP450) catalyzes the formation of an unstable intermediate and highly reactive substrate, known as AFB_1_-8,9-epoxide, as a pivotal event in AFB_1_-induced toxicity mechanisms. This intermediate compound plays the main role in carcinogenic and other toxicities related to AFB_1_ [21,22,23,24]. Additionally, AFB_1_ is associated with immunotoxicity, oxidative stress, and epigenetic changes, such as DNA methylation and RNA alterations, among other effects that potentially contribute to hepatocellular carcinoma (HCC) [13]. In utero, AFB_1_ exposure impacts the offspring’s DNA methylation, highlighting the need for further research to understand the underlying epigenetic mechanisms [5]. Amongst FBs, FB_1_ has been shown to induce toxicity, like neurotoxic, teratogenic, hepatotoxic, and carcinogenic effects in animals [25]. Exposure of pigs to FB_1_ leads to the development of pig-specific clinical dysfunction, namely, pulmonary edema, which is associated with higher pulmonary capillary hydrostatic pressure [19]. Exposure to FB_1_ in pigs also results in damage to the hepatic, cardiovascular, gastrointestinal, and immune systems. Due to the nonsteroidal osteogenic structure, ZEN mimics natural hormones, leading to reproductive issues in animals by reducing estrogen activity and altering the associated metabolic pathways [26]. For example, in pigs, short-term exposure in the first reproduction cycle led to an elevated return to estrus rates, abortions, and hyperestrogenism symptoms in newborn piglets [27]. This exposure also resulted in ovarian follicle atresia, apoptotic-like changes in granule cells, and increased cell proliferation in the uterus and oviduct [26,27].

In the present experiment, no significant differences were observed in the serum biochemical parameters. These findings are consistent with the previously published studies [28,29,30]. Of note, these studies have reported that the abovementioned variables may not be satisfactory biomarkers that could indicate poisoning in pigs exposed to low levels of mycotoxins and/or for a short period of time.

The mycotoxin mixture containing AFs, FBs, and ZEN increased the relative weight of the liver from pigs, also determining hepatocellular dysplasia in this organ. Additionally, these animals also presented interstitial pneumonitis, renal glomerular atrophy, and reduced oocyte number in the ovaries. However, the addition of 3.0 kg/ton OMC or MMDA (1.5 or 3.0 kg/ton) had a positive effect on the relative weight of the liver after 42 days of dietary exposure to AFs, FBs, and ZEN. A similar protective effect of OMC or MMDA was observed regarding the relative weight of the uterus. Moreover, OMC and MMDA treatments were able to reduce the incidence of hepatocellular dysplasia, renal glomerular atrophy, and interstitial pneumonitis. Curiously, the addition of 1.5 kg/ton MMDA increased the oocyte number in the ovaries. On the other hand, pigs supplemented with 3.0 kg/ton OMC or MMDA exhibited mild interstitial inflammatory infiltrate in the kidneys, and the MMDA-treated animals developed hepatitis.

Residual levels of AFs (including AFM_1_) were detected in piglets’ livers only in treatment B (BD + AFs + FBs + ZEN), while the concentrations of FBs and ZEN and their derivative metabolites (α- and β-ZEL) were found to be reduced in the remaining treatments C, D, E, and F, which were statistically lower than treatment B. These results indicate that the treatments with adsorbents were highly protective against the given mycotoxins in piglets’ livers. The protective effect of MMDA was previously demonstrated in an in vivo study [19] with weaned pigs exposed to dietary ZEN and T-2 toxins, indicating a dose-dependent reduction in the residual levels of ZEN and T-2 in the liver, with the best inclusion of this adsorbent at 3.0 kg/ton. Similarly, the use of a purified clay mineral based on bentonite in pig diets during a 35-day trial also increased feed intake and weight gain [31]. According to Raj et al. [16], the use of OMC in the feed of broilers exposed to AFB_1_ and OTA also improved feed conversion and gain in the average body weight, which agrees with the data reported here.

When kidneys were examined for the determination and quantification of residual levels of AFs, FBs, ZEN, and α- and β-ZEL, all treatments also had reduced levels of mycotoxins, compared with treatment B. Along with the evaluation of mycotoxins, AFs reduction by the adsorbents was found to be highly effective in treatments C and D, followed by treatments E and F. For FB_1_ and ZEN, treatments E and F were more effective than the other treatments. These findings indicate that the applied adsorbents, OMC and MMDA, are highly effective against the AFs, also alleviating the effects of FBs and ZEN. Similar efficacy for MMDA was assessed in weaned pigs exposed to dietary ZEN and T-2, in which the residual levels of these toxins decreased significantly, compared with controls receiving only the mycotoxins [12]. In line with the outcomes observed for OMC, a study found that the application of this adsorbent effectively reduced the residual levels of AFB_1_ and OTA in broilers, also improving specific biochemical markers associated with liver health and performance metrics [16].

## 4. Conclusions

In this experiment, the effectiveness of two commercial adsorbents, OMC and MMDA, to reduce the combined adverse effects of dietary AFs, FBs, and ZEN and the residual concentrations of mycotoxin metabolites in the liver and kidney was evaluated in crossbred female pigs. After a 42-day dietary exposure, the mycotoxin cocktail decreased the body weight gain and increased the size of the vulva and the relative weights of the liver and uterus. The mycotoxin mixture also induced moderate histopathological changes in the liver, kidneys, lungs, and ovarium, although no effect was observed in the serum biochemistry parameters of the intoxicated animals. Both OMC and MMDA adsorbents ameliorated the toxic effects and significantly reduced the residual levels of mycotoxins in the liver and kidneys. Notably, OMC supplementation was able to reduce the initiation of liver carcinogenesis without causing hepatotoxic side effects. These findings underscore the effectiveness of OMC and MMDA for mitigating the adverse effects of dietary mycotoxins in piglets, with prospects for improving mycotoxin management strategies in pig-farming operations.

## 5. Materials and Methods

### 5.1. Animals, Diets, and Experimental Design

The experimental work was evaluated and approved by the Animal Ethics Committee of the Institute of Animal Science and Pastures of Nova Odessa (protocol nº 326-2021). Twenty-four crossbred female piglets (21 days) were purchased from a commercial breeding center, allocated in individual cages, and allowed ad libitum access to feed and water. The health status of the animals was assessed by clinical examination upon arriving in the experimental facility, and at 7-day intervals during the entire experimental period, by a qualified veterinarian. After 14 days of adaptation period, the animals were randomly assigned into 6 experimental groups of 4 pigs each and were submitted during 42 days to the treatments summarized in Table 7. The basal diet (BD), based on a corn and soybean meal-type diet, was formulated to meet the nutritional requirements of growing pigs, as recommended by Grenier et al. [32]. Mycotoxin’s culture materials containing Afs (sum of AFB_1_, AFB_2_, AFG_1_, and AFG_2_) [14], FBs and ZEN, along with the commercial adsorbents (Minazel Plus^®^, OMC, and MycoRaid^®^, MMDA) were added to the BD and mixed in a horizontal/helicoidal mixer for 15 min to achieve the targeted concentration of the mycotoxins. The aflatoxins (AFB_1_, AFB_2_, AFG_1_, and AFG_2_), fumonisins (FB_1_ and FB_2_), and ZEN concentrations were determined by an in-house validated liquid chromatography coupled with tandem mass spectrometry [33], as displayed in Table 7. In addition, all diets were screened by using the same analytical method and found to be free of, or with nondetectable levels, of ochratoxin A (limit of detection, LOD: 0.5 μg/kg) and deoxynivalenol (LOD: 6.1 μg/kg). The animals were weighed at baseline, and at 7-day intervals throughout the experiment. The piglets were also monitored daily for any sign of AFs, FBs, or ZEN toxicity. Feed consumption was measured weekly to calculate the feed conversion (FC).

### 5.2. Sample Collection, Biochemical and Histological Analyses

Blood samples were collected at the beginning and at 14 d intervals throughout the experiment via jugular venipuncture in an evacuated blood-collection system in serum separator clot activator tubes Vacuette^®^ (Greiner Bio-one, Kremsmunster, Austria). Serum samples were split into two aliquots, one immediately used for serum biochemistry determinations and the other stored at –20 ℃ for further possible analysis of mycotoxin biomarkers. TP, ALB, AST, ALT, and ALP were measured using an automated biochemical analyzer. Results for AST, TP, and ALB were expressed as g/dL, while ALT and ALP data were displayed as international units (IU)/L.

At the end of the trial, piglets were subjected to electrical stunning and euthanized by exsanguination. The liver, lungs, kidneys, uterus, and ovaries were separated for evaluation. After weighing, these samples were fixed in a 10 kg/ton buffered formalin. Vulvar measurements (height, width, and length) were performed immediately after euthanasia [12]. Graded alcohol was used for the dehydration of the tissue samples, followed by cleaning with xylene, and then embedded in a liquid paraffin. A 5 µm section was stained with hematoxylin–eosin for a descriptive and semiquantitative histopathological analysis in each organ evaluated [10].

### 5.3. Determination of Mycotoxin Residues in Liver and Kidneys

Duplicate samples of 1 g of ground tissues of the liver and kidney were extracted in acetonitrile: water: acetic acid (79:20:1), as described by Cao et al. [34] and summarized in Appendix A. AFB_1_, AFB_2_, AFM_1_, FB_1_, FB_2_, ZEN, α-zearalenol (α-ZOL), and β-zearalenol (β-ZOL) concentrations in the final extracts were determined using a Waters Acquity I-Class ultraperformance liquid chromatographic (UPLC) system (Waters, Milford, MA, USA) equipped with a BEH C18 column (2.1 × 50 mm, 1.7 μm) and coupled to a Xevo TQ-S mass spectrometer (Waters, Milford, MA, USA). The mass spectrometer (MS) was operated in multireaction monitoring (MRM) using electrospray ionization in positive and negative ion modes, with the main parameters as described in Appendix A. Mycotoxin standard solutions and calibration curves were prepared using a work solution with mixed mycotoxins prepared in water: acetonitrile (50:50), containing AFM_1_, AFB_1_, AFB_2_, AFG_1_, AFG_2_, FB_1_, FB_2_, ZEN, α-ZEL, and β-ZEL at 100 ng/mL. This solution was used to prepare five matrix-matched calibration standards at the range levels expressed in Table 3. Additionally, isotopically labeled internal standards (IS) of [^13^C_17_]-AFB_1_ (St. Louis, MO, USA), [^13^C34]-FB_1_ and [^13^C_18_]-ZEN (Biopure, Tulln, Austria) were also prepared in water: acetonitrile (50:50), and added to each sample prior extraction, to reach the concentration of 100 ng/mL for each IS.

Five 5 μL of the extracts and standards were injected using gradient elution in a mobile phase made up of water (eluent A) and acetonitrile (eluent B), both containing 5 mM ammonium acetate and 0.1% acetic acid and kept at 0.6 mL/min, as described elsewhere [33]. The total chromatography run for each sample was 10 min. Limits of detection (LOD) and quantification (LOQ) were determined considering signal-to-noise ratios of 1:3 and 1:10, respectively, and are displayed in Appendix A. The analytical results were based on a standard calibration with added IS, which compensated for both recovery losses and matrix effects.

### 5.4. Statistical Analysis

Data were submitted to analysis of variance (ANOVA) using the PROC GLM of the SAS for Windows program, version 9.4. For multiple comparisons between treatments, the Tukey test was performed. All statements of significance were based on the 0.05 level of probability.

## Figures and Tables

**Figure 1 toxins-15-00629-f001:**
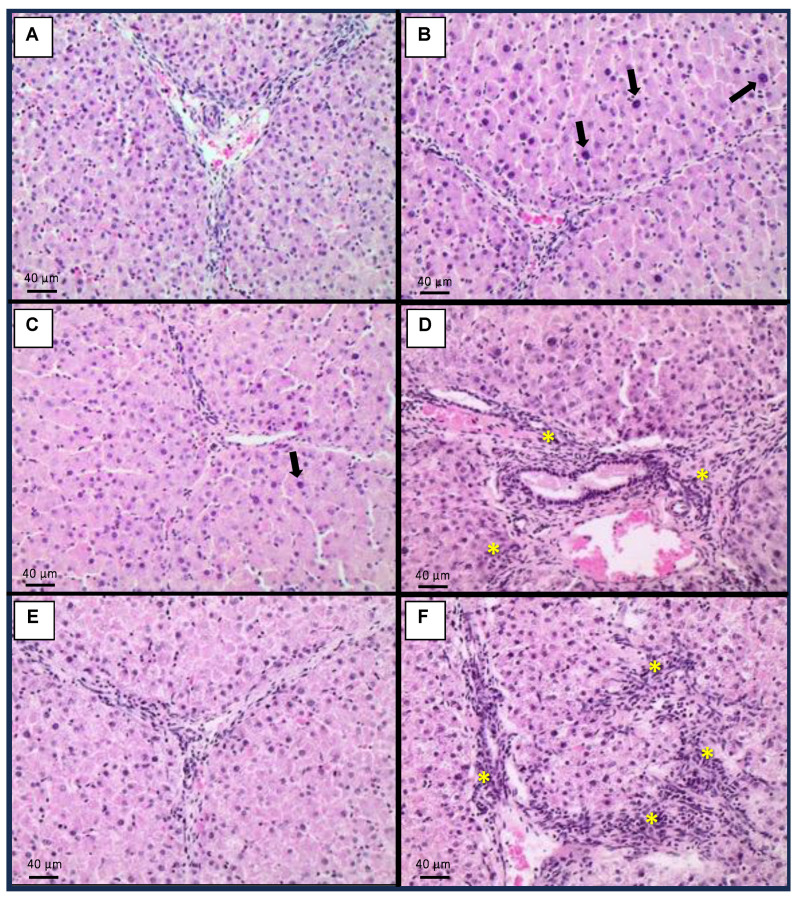
Histopathological findings in the liver. (**A**): Control, only basal diet (BD); (**B**): BD + aflatoxins (AFs) + fumonisins (FBs) + zearalenone (ZEN); (**C**): BD + AFs + FBs + ZEN + 1.5 kg/ton organically modified clinoptilolite (OMC); (**D**): BD + AFs + FBs + ZEN + 1.5 kg/ton multicomponent mycotoxin detoxifying agent (MMDA); (**E**): BD + AFs + FBs + ZEN + 3.0 kg/ton OMC; (**F**): BD + AFs + FBs + ZEN + 3.0 kg/ton MMDA. Arrows indicate moderate liver dysplasia. Asterisks indicate moderate hepatitis.

**Figure 2 toxins-15-00629-f002:**
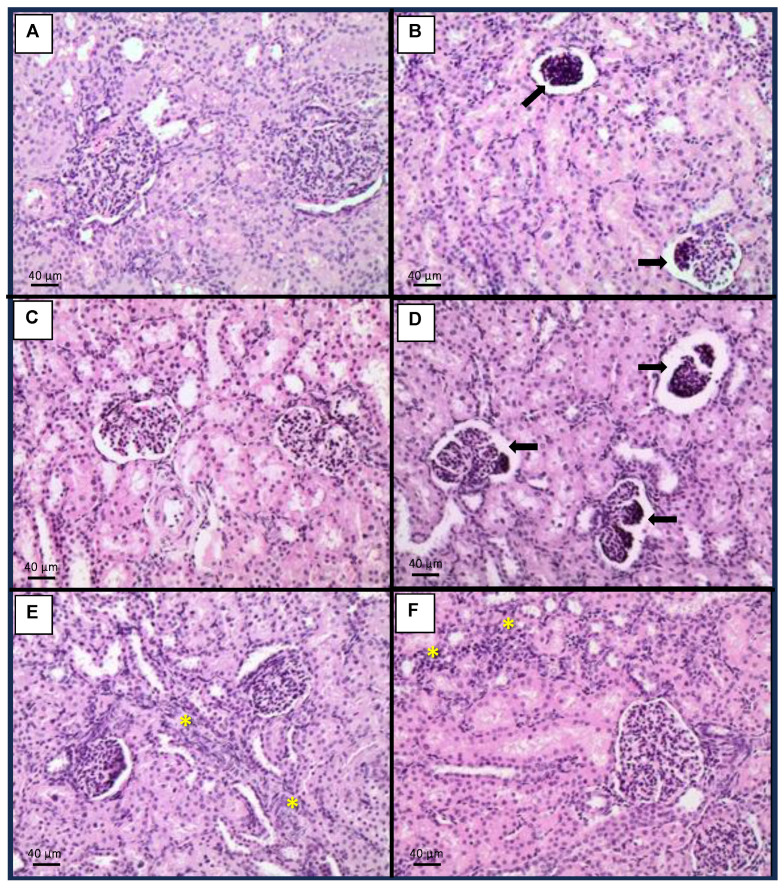
Histopathological findings in the kidneys. (**A**): Control, only basal diet (BD); (**B**): BD + aflatoxins (AFs) + fumonisins (FBs) + zearalenone (ZEN); (**C**): BD + AFs + FBs + ZEN + 1.5 kg/ton organically modified clinoptilolite (OMC); (**D**): BD + AFs + FBs + ZEN + 1.5 kg/ton multicomponent mycotoxin detoxifying agent (MMDA); (**E**): BD + AFs + FBs + ZEN + 3.0 kg/ton OMC; (**F**): BD + AFs + FBs + ZEN + 3.0 kg/ton MMDA. Arrows indicate renal glomerular atrophy. Asterisks indicate mild renal interstitial inflammation.

**Figure 3 toxins-15-00629-f003:**
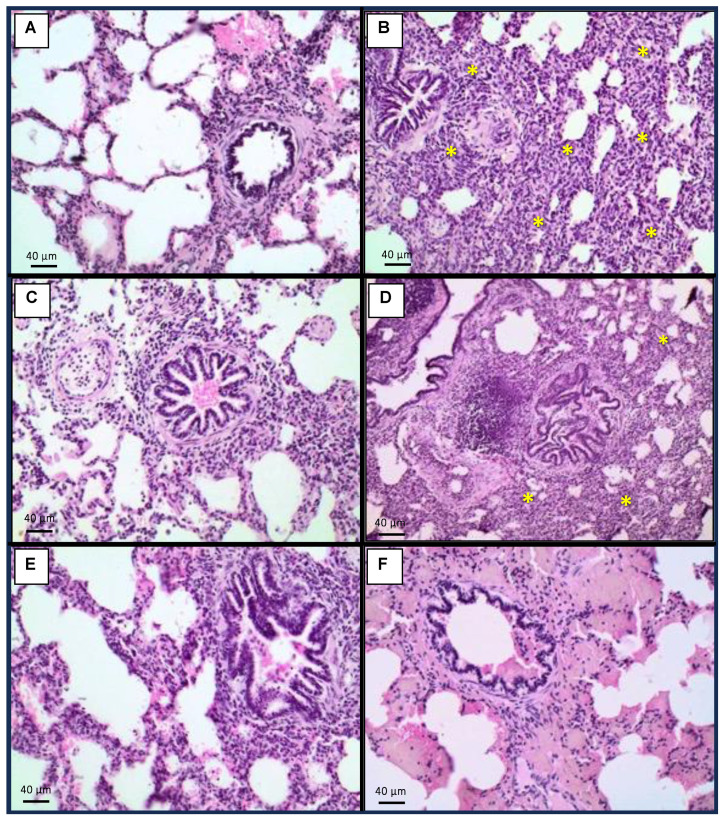
Histopathological findings in the lungs. (**A**): Control, only basal diet (BD); (**B**): BD + aflatoxins (AFs) + fumonisins (FBs) + zearalenone (ZEN); (**C**): BD + AFs + FBs + ZEN + 1.5 kg/ton organically modified clinoptilolite (OMC); (**D**): BD + AFs + FBs + ZEN + 1.5 kg/ton multicomponent mycotoxin detoxifying agent (MMDA); (**E**): BD + AFs + FBs + ZEN + 3.0 kg/ton OMC; (**F**): BD + AFs + FBs + ZEN + 3.0 kg/ton MMDA. Asterisks indicate moderate interstitial pneumonitis.

**Figure 4 toxins-15-00629-f004:**
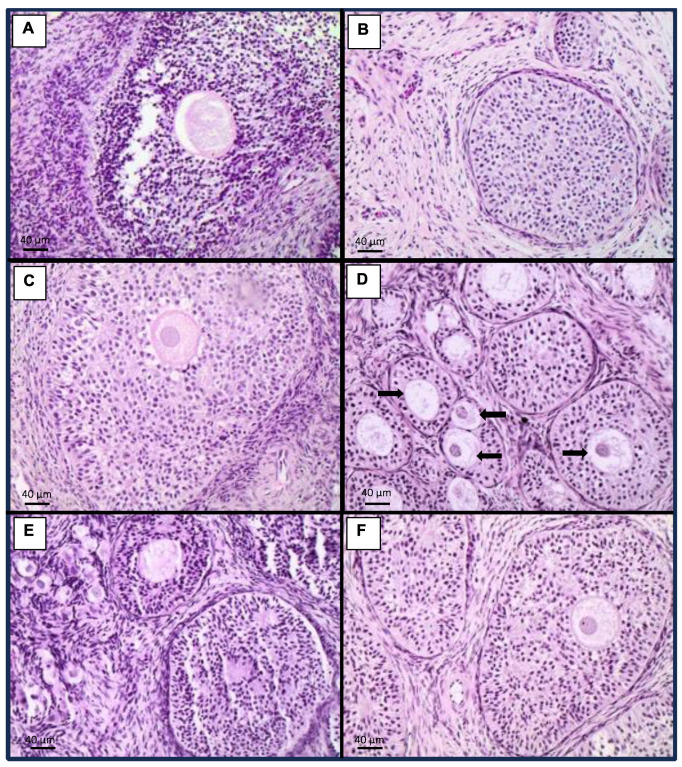
Histopathological findings in the ovaries. (**A**): Control, only basal diet (BD); (**B**): BD + aflatoxins (AFs) + fumonisins (FBs) + zearalenone (ZEN); (**C**): BD + AFs + FBs + ZEN + 1.5 kg/ton organically modified clinoptilolite (OMC); (**D**): BD + AFs + FBs + ZEN + 1.5 kg/ton multicomponent mycotoxin detoxifying agent (MMDA); (**E**): BD + AFs + FBs + ZEN + 3.0 kg/ton OMC; (**F**): BD + AFs + FBs + ZEN + 3.0 kg/ton MMDA. Arrows indicate increased oocyte number.

**Table 1 toxins-15-00629-t001:** Efficacy of commercial adsorbents to ameliorate the toxic effects of combined dietary exposure to mycotoxin mixture on the performance of piglets after 42 days of intoxication ^1^.

Treatment ^2^	Body-Weight Gain (kg)	Feed Consumption (kg)	Feed: Gain
A	29.8 ± 0.97 ^a^	8.12 ± 0.59 ^a^	0.21 ± 0.01 ^a^
B	24.4 ± 2.15 ^b^	6,74 ± 1.24 ^a^	0.19 ± 0.02 ^a^
C	26.7 ± 3.23 ^ab^	6.39 ± 1.22 ^a^	0.20 ± 0.03 ^a^
D	30.2 ± 1.73 ^a^	6.63 ± 1.02 ^a^	0.19 ± 0.02 ^a^
E	28.5 ± 2.08 ^a^	6.22 ± 0.27 ^a^	0.20 ± 0.03 ^a^
F	27.9 ± 1.19 ^a^	6.67 ± 0.57 ^a^	0.19 ± 0.02 ^a^

^1^ Data are means ± SD of 4 female piglets per treatment. ^2^ A: Control, only basal diet (BD); B: BD + aflatoxins (AFs) + fumonisins (FBs) + zearalenone (ZEN); C: BD + AFs + FBs + ZEN + 1.5 kg/ton organically modified clinoptilolite (OMC); D: BD + AFs + FBs + ZEN + 3.0 kg/ton OMC; E: BD + AFs + FBs + ZEN + 1.5 kg/ton multicomponent mycotoxin detoxifying agent (MMDA); F: BD + AFs + FBs + ZEN + 3.0 kg/ton MMDA. ^a,b^ Values within each column with no common superscript differ significantly (*p* < 0.05).

**Table 2 toxins-15-00629-t002:** Efficacy of commercial adsorbents to ameliorate the toxic effects of combined dietary exposure to mycotoxin mixture on serum biochemistry of piglets during 42 days of intoxication ^1^.

Treatment ^2^	TP (g/dL)	ALB (g/dL)	AST (g/dL)	ALT (IU/L)	ALP (IU/L)
A	5.05 ± 0.78 ^a^	3.13 ± 0.59 ^a^	31.5 ± 12.2 ^a^	58.6 ± 28.5 ^a^	392 ± 207 ^a^
B	4.76 ± 0.36 ^a^	3.00 ± 0.38 ^a^	35.8 ± 8.24 ^a^	60.8 ± 27.8 ^a^	367 ± 186 ^a^
C	5.01 ± 0.86 ^a^	3.37 ± 0.64 ^a^	32.1 ± 16.0 ^a^	53.1 ± 20.7 ^a^	378 ± 154 ^a^
D	4.95 ± 0.94 ^a^	3.35 ± 0.62 ^a^	34.5 ± 11.5 ^a^	58.0 ± 24.0 ^a^	419 ± 180 ^a^
E	4.80 ± 0.64 ^a^	3.42 ± 0.50 ^a^	33.4 ± 10.7 ^a^	48.6 ± 15.0 ^a^	405 ± 184 ^a^
F	5.06 ± 1.21 ^a^	3.11 ± 0.53 ^a^	24.1 ± 8.64 ^a^	45.3 ± 13.1 ^a^	379 ± 121 ^a^

^1^ Data are means ± SD of 4 female piglets per treatment, measured at 14-days intervals from day 1 to 42 of intoxication. ^2^ A: Control, only basal diet (BD); B: BD + aflatoxins (AFs) + fumonisins (FBs) + zearalenone (ZEN); C: BD + AFs + FBs + ZEN + 1.5 kg/ton organically modified clinoptilolite (OMC); D: BD + AFs + FBs + ZEN + 3.0 kg/ton OMC; E: BD + AFs + FBs + ZEN + 1.5 kg/ton multicomponent mycotoxin detoxifying agent (MMDA); F: BD + AFs + FBs + ZEN + 3.0 kg/ton MMDA. TP: Total protein; ALB: Albumin; AST: Serum aspartate aminotransferase; ALT: Alanine aminotransferase; ALP: Alkaline phosphatase. ^a^ No significant differences (*p* > 0.05) were found in the mean values.

**Table 3 toxins-15-00629-t003:** Efficacy of commercial adsorbents to ameliorate the toxic effects of combined dietary exposure to mycotoxin mixture on relative organ weights (g/kg body weight) of piglets after 42 days of intoxication ^1^.

Treatment ^2^	Liver	Kidneys	Uterus	Ovarium	Lungs
A	2.36 ± 0.28 ^b^	0.51 ± 0.04 ^a^	0.12 ± 0.01 ^b^	0.01 ± 0.00 ^a^	1.27 ± 0.14 ^a^
B	3.43 ± 0.41 ^a^	0.51 ± 0.05 ^a^	0.22 ± 0.09 ^a^	0.01 ± 0.01 ^a^	1.24 ± 0.21 ^a^
C	3.61 ± 0.69 ^a^	0.54 ± 0.04 ^a^	0.19 ± 0.05 ^ab^	0.00 ± 0.00 ^a^	1.40 ± 0.21 ^a^
D	2.89 ± 0.21 ^ab^	0.49 ± 0.03 ^a^	0.15 ± 0.03 ^b^	0.00 ± 0.00 ^a^	1.23 ± 0.14 ^a^
E	3.13 ± 0.60 ^ab^	0.54 ± 0.10 ^a^	0.19 ± 0.05 ^ab^	0.01 ± 0.02 ^a^	1.55 ± 0.82 ^a^
F	3.03 ± 0.36 ^ab^	0.47 ± 0.02 ^a^	0.16 ± 0.02 ^b^	0.01 ± 0.01 ^a^	1.34 ± 0.53 ^a^

^1^ Data are means ± SD of 4 female piglets per treatment. ^2^ A: Control, only basal diet (BD); B: BD + aflatoxins (AFs) + fumonisins (FBs) + zearalenone (ZEN); C: BD + AFs + FBs + ZEN + 1.5 kg/ton organically modified clinoptilolite (OMC); D: BD + AFs + FBs + ZEN + 3.0 kg/ton OMC; E: BD + AFs + FBs + ZEN + 1.5 kg/ton multicomponent mycotoxin detoxifying agent (MMDA); F: BD + AFs + FBs + ZEN + 3.0 kg/ton MMDA. ^a,b^ Values within each column with no common superscript differ significantly (*p* < 0.05).

**Table 4 toxins-15-00629-t004:** Efficacy of commercial adsorbents to ameliorate the toxic effects of combined dietary exposure to mycotoxin mixture on size measurements (cm) of vulvas from piglets after 42 days of intoxication ^1^.

Treatment ^2^	Width	Height	Length
A	2.13 ± 0.40 ^c^	2.37 ± 0.35 ^c^	1.72 ± 0.25 ^c^
B	2.76 ± 0.81 ^a^	3.54 ± 0.52 ^a^	2.34 ± 0.31 ^a^
C	2.50 ± 0.40 ^ab^	3.19 ± 0.14 ^ab^	1.90 ± 0.33 ^bc^
D	2.49 ± 0.43 ^ab^	3.20 ± 0.39 ^ab^	2.13 ± 0.40 ^b^
E	2.31 ± 0.23 ^b^	2.89 ± 0.45 ^b^	2.09 ± 0.25 ^b^
F	2.44 ± 0.18 ^ab^	2.96 ± 0.29 ^b^	2.03 ± 0.25 ^bc^

^1^ Data are means ± SD of 4 female piglets per treatment. ^2^ A: Control, only basal diet (BD); B: BD + aflatoxins (AFs) + fumonisins (FBs) + zearalenone (ZEN); C: BD + AFs + FBs + ZEN + 1.5 kg/ton organically modified clinoptilolite (OMC); D: BD + AFs + FBs + ZEN + 3.0 kg/ton OMC; E: BD + AFs + FBs + ZEN + 1.5 kg/ton multicomponent mycotoxin detoxifying agent (MMDA); F: BD + AFs + FBs + ZEN + 3.0 kg/ton MMDA. ^a–c^ Values within each column with no common superscript differ significantly (*p* < 0.05).

**Table 5 toxins-15-00629-t005:** Efficacy of commercial adsorbents to ameliorate the toxic effects of combined dietary exposure to mycotoxin mixture on mycotoxin residues in the liver of piglets after 42 days of intoxication ^1^.

Treatment ^2^	AFM_1_ (µg/kg)	AFs ^3^ (µg/kg)	FBs ^4^ (µg/kg)	ZEN (µg/kg)	α-ZEL (µg/kg)	β-ZEL (µg/kg)
A	<LOQ	<LOQ	<LOQ	<LOQ	<LOQ	<LOQ
B	0.92 ± 0.07	3.99 ± 0.22	2.59 ± 0.97 ^a^	23.3 ± 3.7 ^a^	<LOQ	<LOQ
C	<LOQ	<LOQ	1.67 ± 0.10 ^c^	19.3 ± 3.2 ^b^	<LOQ	<LOQ
D	<LOQ	<LOQ	1.99 ± 0.78 ^b^	16.1 ± 3.6 ^c^	<LOQ	<LOQ
E	<LOQ	<LOQ	1.52 ± 0.88 ^c^	15.7 ± 2.4 ^c^	<LOQ	<LOQ
F	<LOQ	<LOQ	1.65 ± 0.59 ^c^	10.5 ± 1.7 ^d^	<LOQ	<LOQ

^1^ Data are means ± SD of 4 female piglets per treatment. ^2^ A: Control, only basal diet (BD); B: BD + aflatoxins (AFs) + fumonisins (FBs) + zearalenone (ZEN); C: BD + AFs + FBs + ZEN + 1.5 kg/ton organically modified clinoptilolite (OMC); D: BD + AFs + FBs + ZEN + 3.0 kg/ton OMC; E: BD + AFs + FBs + ZEN + 1.5 kg/ton multicomponent mycotoxin detoxifying agent (MMDA); F: BD + AFs + FBs + ZEN + 3.0 kg/ton MMDA. ^3^ Sum of AFB_1_ + AFB_2_ + AFG_1_ + AFG_2_. ^4^ Sum of FB_1_ + FB_2_. ^a–d^ Values within each column with no common superscript differ significantly (*p* < 0.05). LOQ: Limit of quantification (see Appendix A for LOQ values for each mycotoxin); ZEL: zearalenol.

**Table 6 toxins-15-00629-t006:** Efficacy of commercial adsorbents to ameliorate the toxic effects of combined dietary exposure to mycotoxin mixture on mycotoxin residues in piglet’s kidney after 42 days of intoxication ^1^.

Treatment ^2^	AFM_1_ (µg/kg)	Afs ^3^ (µg/kg)	FBs ^4^ (µg/kg)	ZEN (µg/kg)	α-ZEL (µg/kg)	β-ZEL (µg/kg)
A	<LOQ	<LOQ	<LOQ	<LOQ	<LOQ	<LOQ
B	2.63 ± 0.31 ^a^	9.13 ± 0.25 ^a^	4.32 ± 0.65 ^a^	41.6 ± 5.7 ^a^	<LOQ	<LOQ
C	<LOQ	<LOQ	2.98 ± 1.11 ^b^	34.3 ± 3.3 ^b^	<LOQ	<LOQ
D	<LOQ	<LOQ	3.25 ± 1.56 ^b^	30.5 ± 4.2 ^c^	<LOQ	<LOQ
E	1.06 ± 0.20 ^b^	4.25 ± 0.53 ^b^	2.77 ± 1.02 ^c^	23.1 ± 2.9 ^d^	<LOQ	<LOQ
F	0.80 ± 0.11 ^b^	1.92 ± 0.24 ^c^	2.63 ± 1.32 ^c^	17.4 ± 2.1 ^d^	<LOQ	<LOQ

^1^ Data are means ± SD of 4 female piglets per treatment. ^2^ A: Control, only basal diet (BD); B: BD + aflatoxins (AFs) + fumonisins (FBs) + zearalenone (ZEN); C: BD + AFs + FBs + ZEN + 1.5 kg/ton organically modified clinoptilolite (OMC); D: BD + AFs + FBs + ZEN + 3.0 kg/ton OMC; E: BD + AFs + FBs + ZEN + 1.5 kg/ton multicomponent mycotoxin detoxifying agent (MMDA); F: BD + AFs + FBs + ZEN + 3.0 kg/ton MMDA. ^3^ Sum of AFB_1_ + AFB_2_ + AFG_1_ + AFG_2_. ^4^ Sum of FB_1_ + FB_2_. ^a–d^ Values within each column with no common superscript differ significantly (*p* < 0.05). LOQ: Limit of quantification (see Appendix A for LOQ values for each mycotoxin); ZEL: zearalenol.

**Table 7 toxins-15-00629-t007:** Dietary treatments and respective concentrations of mycotoxins in experimental feeds ^1^.

Treatment ^2^	AFB_1_ (μg/kg)	AFB_2_ (μg/kg)	AFG_1_ (μg/kg)	AFG_1_ (μg/kg)	FB_1_ (μg/kg)	FB_2_ (μg/kg)	ZEN (μg/kg)
A	<LOQ	<LOQ	<LOQ	<LOQ	<LOQ	<LOQ	<LOQ
B	310 ± 40.2	43.1 ± 9.80	132 ± 23.4	11.3 ± 12.7	935 ± 105	210 ± 98.2	847 ± 43,5
C	284 ± 32.3	34.0 ± 10.2	123 ± 22.1	14.8 ± 9.60	847 ± 98.5	232 ± 101	985 ± 73.4
D	275 ± 23.0	32.8 ± 9.65	113 ± 30.2	10.3 ± 9.6	997 ± 99.7	324 ± 97.3	943 ± 64.7
E	314 ± 95.2	33.5 ± 9.76	121 ± 22.5	9.88 ± 9.66	955 ± 102	298 ± 100	939 ± 62.2
F	312 ± 98.5	38.6 ± 10.3	134 ± 42.3	11.4 ± 9.75	998 ± 98.8	196 ± 98.5	995 ± 82.3

^1^ Mean ± SD of five samples analyzed individually. ^2^ A: Control, only basal diet (BD); B: BD + aflatoxins (AFs) + fumonisins (FBs) + zearalenone (ZEN); C: BD + AFs + FBs + ZEN + 1.5 kg/ton organically modified clinoptilolite (OMC); D: BD + AFs + FBs + ZEN + 3.0 kg/ton OMC; E: BD + AFs + FBs + ZEN + 1.5 kg/ton multicomponent mycotoxin detoxifying agent (MMDA); F: BD + AFs + FBs + ZEN + 3.0 kg/ton MMDA. LOQ: Limit of quantification (1.0 μg/kg for each aflatoxin, 2.5 μg/kg for each fumonisin, and 0.6 μg/kg for zearalenone).

## Data Availability

The data related to this work is within the manuscript.

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
