# Peer review of "Efficacy of Two Commercially Available Adsorbents to Reduce the Combined Toxic Effects of Dietary Aflatoxins, Fumonisins, and Zearalenone and Their Residues in the Tissues of Weaned Pigs"

_toxins, 2023, doi:10.3390/toxins15110629_

Round 1

Reviewer 1 Report

Comments and Suggestions for Authors

The author should make some major revisions according to my recommendations.

·        The sentences in abstract line 13-14 repeat the same results, please revise them.

·        Conclusion and suggestion are missing from the abstract. Revise it.

·        Aflatoxins are various poisonous carcinogens and mutagens that are produced by certain molds. The major types of AFB1 and AFB2 are their metabolites. The author should clearly mention which metabolite was administered. Mention in the title, abstract, method, and results.

·        Aflatoxin dose was administered by any reference study to induce the toxicity in pigs? Mention in method

·        Table 1 subheading treatmens ? Check spelling.

·        Detailed information on Dietary treatments and their respective concentrations of mycotoxins in experimental feeds is provided in Table 1, and the author has provided tables from 2-6 with the same repetition. My recommendation is for the author to modify the tables and write the description for example, Group 1-6 treatments or group A-E names or numbers only rather than repetition of lengthy group names.

·        The results of P, ALB, AST, ALT or ALP were expressed in g/Dl or IU/L. This detail is missing from the material and method section. Please provide this detail in the method.

·        Table 2 title is very lengthy, please revise it.

·        Line 102-14 results are unclear. Please revise them for clarity.

·        Revise the titles of all tables. The titles should be concise.

·        Histopathological results of figure, I suggest figure 1 should be separated for each organ results, the quality of the figure is poor, as a recommended quality should be at least 100 um. Separate the results of all organs and write them in different subheadings.

·        The conclusion of the study is unclear and poorly written. Give a brief summary of your paper and discuss whether or not it addresses your research questions.

·        Overall manuscript required moderate editing of English language.

Comments on the Quality of English Language

    Overall manuscript required moderate editing of English language.

Author Response

Answers to Reviewers:

The authors thank the constructive comments and suggestions of the Reviewers for improving the manuscript. We have addressed all the concerns regarding the manuscript and included below a point-by-point response to the Reviewers. In addition, the changes done in the text were highlighted in yellow in the new, revised version of the manuscript.

Reviewer #1:

The sentences in abstract line 13-14 repeat the same results, please revise them.

Answer: Thanks for the comments. The repeated sentence in the Abstract was excluded, as indicated.

Conclusion and suggestion are missing from the abstract. Revise it.

Answer: Conclusion and suggestion statements were amended in the revised manuscript.

Aflatoxins are various poisonous carcinogens and mutagens that are produced by certain molds. The major types of AFB1 and AFB2 are their metabolites. The author should clearly mention which metabolite was administered. Mention in the title, abstract, method, and results.

Answer: The culture material administered to pigs was a mixture of aflatoxin metabolites (AFs: sum of AFB1, AFB2, AFG1 and AFG2), as detailed in Table 7 (former Table 1 in the original version of the manuscript). This information was amended in the abstract, Material and methods, and Results sections, as suggested.

Aflatoxin dose was administered by any reference study to induce the toxicity in pigs? Mention in method.

Answer: Yes, the aflatoxin administration was based on the procedures as described by Neef et al. (2013), as amended in the revised manuscript (reference #15).

Table 1 subheading treatmens ? Check spelling.

Answer: Done.

Detailed information on Dietary treatments and their respective concentrations of mycotoxins in experimental feeds is provided in Table 1, and the author has provided tables from 2-6 with the same repetition. My recommendation is for the author to modify the tables and write the description for example, Group 1-6 treatments or group A-E names or numbers only rather than repetition of lengthy group names.

Answer: Thanks for the suggestion. The treatment names in all tables and throughout the text were revised accordingly.

The results of P, ALB, AST, ALT or ALP were expressed in g/Dl or IU/L. This detail is missing from the material and method section. Please provide this detail in the method.

Answer: The information requested was amended in the 1st paragraph of section 5.2. “Sample Collection, Biochemical and Histological Analyses”.

Table 2 title is very lengthy, please revise it.

Answer: The titles of Table 2 was revised, as suggested.

Line 102-14 results are unclear. Please revise them for clarity.

Answer: Those lines were rephrased for clarity.

Revise the titles of all tables. The titles should be concise.

Answer: The titles of all tables were summarized accordingly.

Histopathological results of figure, I suggest figure 1 should be separated for each organ results, the quality of the figure is poor, as a recommended quality should be at least 100 um. Separate the results of all organs and write them in different subheadings.

Answer: Thanks for the suggestion. Figure 1 was separated in 4 new figures showing the results for each organ. A new section (2.4.“Histopathology”) was included to proper describe the histopathological results for each evaluated organ.   

The conclusion of the study is unclear and poorly written. Give a brief summary of your paper and discuss whether or not it addresses your research questions.

Answer: Conclusion was improved as per the recommendation.

Overall manuscript required moderate editing of English language.

Answer: The text was thoroughly revised by a native English speaker.

Reviewer 2 Report

Comments and Suggestions for Authors

Review report “toxins-2653312

In the manuscript entitled “Efficacy of two commercially available adsorbents to reduce the combined toxic effects of dietary aflatoxins, fumonisins and zearalenone and their residues in tissues of weaned pigs” the authors report on the evaluation of efficacy of two commercial adsorbents as detoxification agents of dietary mycotoxins.

The manuscript is organized in a logical way. The length is appropriate. The abstract and introduction are quite clear and sufficiently reflect the manuscript content.

The treated topic is of great interest and the animal study was carried out is a good and transparent manner. In my opinion, these types of studies should be implemented.

For this reason, I strongly invite the authors to improve this manuscript, in particular requesting the help of a native English speaker for proofreading, because it deserves to be published after several refinements. There are many many typos, grammar imperfections and concordance errors (e.g. line 34,35, 107, 196,197, 212, 242, 251, 329,359, table 4 ……….). It is annoying during the reading. Similarly, several sentences need to be rewritten.

Line 327 and line 359. It should be table 7.

Finally, the section “conclusions” should be implemented. It should present again the objectives of the study, then show the major and most important results.

Based on these comments, I recommend the publication of this manuscript in Toxins as long as these comments will be considered.

Comments on the Quality of English Language

Review report “toxins-2653312”

In the manuscript entitled “Efficacy of two commercially available adsorbents to reduce the combined toxic effects of dietary aflatoxins, fumonisins and zearalenone and their residues in tissues of weaned pigs” the authors report on the evaluation of efficacy of two commercial adsorbents as detoxification agents of dietary mycotoxins.

The manuscript is organized in a logical way. The length is appropriate. The abstract and introduction are quite clear and sufficiently reflect the manuscript content.

The treated topic is of great interest and the animal study was carried out is a good and transparent manner. In my opinion, these types of studies should be implemented.

For this reason, I strongly invite the authors to improve this manuscript, in particular requesting the help of a native English speaker for proofreading, because it deserves to be published after several refinements. There are many many typos, grammar imperfections and concordance errors (e.g. line 34,35, 107, 196,197, 212, 242, 251, 329,359, table 4 ……….). It is annoying during the reading. Similarly, several sentences need to be rewritten.

Line 327 and line 359. It should be table 7.

Finally, the section “conclusions” should be implemented. It should present again the objectives of the study, then show the major and most important results.

Based on these comments, I recommend the publication of this manuscript in Toxins as long as these comments will be considered.

Author Response

Answers to Reviewers:

The authors thank the constructive comments and suggestions of the Reviewers for improving the manuscript. We have addressed all the concerns regarding the manuscript and included below a point-by-point response to the Reviewers. In addition, the changes done in the text were highlighted in yellow in the new, revised version of the manuscript.

Reviewer #2:

The manuscript is organized in a logical way. The length is appropriate. The abstract and introduction are quite clear and sufficiently reflect the manuscript content.

Answer: The authors thank for your comments.

The treated topic is of great interest and the animal study was carried out is a good and transparent manner. In my opinion, these types of studies should be implemented.

Answer: Thanks for your comments.

For this reason, I strongly invite the authors to improve this manuscript, in particular requesting the help of a native English speaker for proofreading, because it deserves to be published after several refinements. There are many many typos, grammar imperfections and concordance errors (e.g. line 34,35, 107, 196,197, 212, 242, 251, 329,359, table 4 ……….). It is annoying during the reading. Similarly, several sentences need to be rewritten.

Answer: All The text was thoroughly revised by a native English speaker.

Line 327 and line 359. It should be table 7.

Answer: Corrected.

Finally, the section “conclusions” should be implemented. It should present again the objectives of the study, then show the major and most important results.

Answer: The Conclusion section was improved accordingly.

Based on these comments, I recommend the publication of this manuscript in Toxins as long as these comments will be considered.

Answer: The authors sincerely hope that all your valuable comments were fully addressed in the new, revised version of the manuscript.

Round 2

Reviewer 1 Report

Comments and Suggestions for Authors

Author revised all my suggestions, I still have few concerns with histopathology results description, the author should describe the results in the result section rather than in the figure caption. Also, clearly mention each figure number after the results of each organ correctly. The caption should be simple to provide only detail description of A, B, C ,, for example A (Control) B, (aflatoxin). …. etc.  Figure 3 is mentioned two times.

Author Response

Answer: Thanks for the comments. The description of histopathology results was transferred from the captions in each figure to the section text, as requested. Therefore, each figure caption was kept with simple information regarding the treatments (A-Control, B-Mycotoxin mixture, etc). Figure 4 (formerly, Figure 3 duplicate) was corrected, as pointed out.